# Formation of Aroma Characteristics in Roasted *Camellia oleifera* Seeds

**DOI:** 10.3390/foods15010087

**Published:** 2025-12-27

**Authors:** Huanling Lan, Xueyuan Lin, Huanhuan Ma, Liuying Lu, Wenxia Liao, Yuting Wang, Yi Chen, Chang Li

**Affiliations:** State Key Laboratory of Food Science and Resources, Nanchang University, Nanchang 330047, China

**Keywords:** camellia oil, aroma precursors, volatile flavor compounds, reaction model systems

## Abstract

*Camellia oleifera* oil (CO) is an important edible oil with excellent nutritional value. Recently, there has been an increasing market demand for oils with distinct flavor profiles. However, the formation mechanisms of characteristic aromas in CO remain unclear. Therefore, this study investigated the effects of roasting (170 °C, 0–30 min) on free amino acids, soluble sugars, and volatile components in camellia seeds and the corresponding oils. To further elucidate the generation mechanisms of flavor compounds in CO, reaction systems simulating the Maillard reaction and lipid oxidation were constructed. The results show strong correlations between volatile compounds and both soluble sugars and free amino acids during roasting. The key flavor precursors identified included arginine, glutamic acid, glycine, histidine, leucine, phenylalanine, and lysine, as well as sucrose and glucose. The simulated systems indicated that the flavor compounds in CO were mainly derived from the Maillard reaction and lipid oxidation, with significant interactions enhancing its unique flavor. This study potentially provides scientific guidance for the production and flavor control of fragrant CO.

## 1. Introduction

Camellia (*Camellia oleifera* Abel.), a member of the Theaceae family, is recognized as one of the four major woody oilseed tree species worldwide [1]. Camellia oil (CO), a high-quality edible oil extracted from camellia seeds, is rich in a variety of unsaturated fatty acids, especially oleic acid (accounting for 75–85%), as well as essential fatty acids such as linoleic acid and α-linolenic acid [2]. Due to its similarity to olive oil in terms of fatty acid profile and bioactive substance content, CO is often referred to as “Oriental olive oil” in the international market [3]. In addition, CO contains a variety of functional components with physiological functions, including squalene, tocopherols, polyphenols, and phytosterols [4]. These functional components exhibit significant bioactivities and may help regulate lipid metabolism, improve cardiovascular and liver function, inhibit tumor cell proliferation, and scavenge free radicals [5,6]. In recent years, with rising living standards, increasing attention has been paid to both the nutritional value and flavor diversity of edible oils. As a result, market demand for strongly flavored CO has grown significantly.

In the production of edible CO, the roasting of camellia seeds is a critical step. Complex chemical changes occur within the seeds during the heating process, generating a variety of aroma substances, which exert a profound influence on the flavor and quality of edible oils [7]. Traditional preheating treatments mainly include roasting, frying, and steaming. Various preheating processes exert distinct influences on the flavor profiles of CO. Flavor compounds are formed in CO during the high-temperature roasting of seeds [8]. Roasting treatment can promote the formation of heterocycles, enhancing the roasted flavor of oils [9]. Some studies have compared different pretreatment methods for camellia seeds, such as microwave heating and frying and found that these methods enhanced the diversity of volatile substances in CO to varying degrees [7]. Currently, roasting is a relatively common preheating technology. During roasting, key reactions including the Maillard reaction, lipid oxidation, and thermal degradation predominantly occur, contributing to the development of distinctive flavor characteristics in the oil [10]. The flavor characteristics of CO are attributable to the synergistic effects of its abundant volatile compounds, which consisted of several short-chain hydrocarbons and a range of polar and non-polar functional groups, including aldehydes, alcohols, ketones, acids, esters, and heterocycles [11,12]. These substances contribute to the unique flavor characteristics of the oil, such as its green, fruity, floral, fatty, and nutty aromas [13,14,15], playing an important role in the overall flavor formation of CO and providing a theoretical basis for optimizing the production of fragrant CO.

The formation of these key volatile flavor compounds is intrinsically linked to the presence and thermal transformation of endogenous flavor precursors, particularly free amino acids and soluble sugars [16]. The Maillard reaction between reducing sugars and amino acids, coupled with the Strecker degradation process, synergistically drives the formation of key nitrogen-containing and oxygen-containing heterocyclic compounds, thereby generating important aroma compounds (e.g., pyrazines and furans) that contribute to baked and nutty flavors. Concurrently, the thermal oxidation of lipids serves as a major pathway, decomposing unsaturated fatty acids into a series of volatile aldehydes and ketones, which impart fatty, green, and creamy notes to the overall aroma profile. Therefore, the specific volatile flavor compound profile of CO depends on the initial composition of these precursors and the processing conditions. However, systematic identification of the specific amino acids and sugars in camellia seeds that act as key aroma precursors, as well as a clear elucidation of their conversion pathways, remains inadequate in the current research.

At present, the primary approaches for investigating aroma formation pathways in oils involve constructing models to analyze the relationships among reaction substrates, intermediates, and final products. Potential aroma formation pathways can be inferred by supplementing reaction substrates to original samples for validation [17]. By analyzing the volatile components in oils, the overall aroma can be characterized, facilitating the identification and prediction of characteristic aroma compounds [18]. The Maillard reaction and lipid oxidation may occur concurrently and interact with each other, with both reaction pathways influencing oil flavor [19,20]. Through the establishment of Maillard reaction, lipid oxidation, and Maillard reaction–lipid oxidation interaction models, it has been demonstrated that sesame lignans can regulate aroma formation and sensory perception of sesame oil via these two pathways [20]. Additionally, high-quality fragrant rapeseed oil with superior flavor can be produced by combining the Maillard reaction and enzymatic digestion [21]. Currently, research on fragrant edible oil mainly focus on fragrant peanut oil, sesame oil, and rapeseed oil. However, to fully understand the aroma formation pathways in fragrant CO, further investigation into the interrelated mechanisms of lipid oxidation, Maillard reactions, and their interactions in aroma compound formation is required.

The present study aimed to systematically identify the key aromatic precursors of CO, reveal the dominant metabolic pathways (Maillard reaction and lipid oxidation) and their interaction mechanisms, underlying the formation of CO’s volatile flavor com-pounds. Under different roasting durations (170 °C, 0–30 min), the dynamic changes in free amino acids, soluble sugars (potential aromatic precursors), and volatile flavor compounds in camellia seeds were analyzed. A partial least squares regression (PLSR) model was used to establish the correlations between amino acids, soluble sugars, and volatile flavor compounds, aiming to clarify the correlations between precursors and volatile flavor compounds and identify key aromatic precursors of CO. A series of simulated reaction systems were constructed, including Maillard reaction models between different amino acids and reducing sugars, a lipid oxidation model incorporating cold-pressed CO, and an interaction model of the lysine–glucose Maillard reaction and lipid oxidation. These model systems were employed to preliminarily deduce the formation mechanisms and pathways of aroma compounds in roasted camellia seeds. This study could lay a theoretical foundation for flavor regulation and optimized production of fragrant CO.

## 2. Materials and Methods

### 2.1. Plant Material and Chemical Reagents

The camellia seeds were collected from Deyiyuan Ecological Agriculture Development Co., Ltd. (Poyang County, Jiangxi Province, China) in 2023. This region features a mid-subtropical monsoon climate with abundant rainfall and ample sunlight. The mature seeds were harvested during the traditional harvest season, then sent to the factory for hulling and drying. The dried seeds were stored at 4 °C cold storage until processing. Unroasted cold-pressed CO was provided by Qiyunshan Camellia Science and Technology Co., Ltd. (Ganzhou, China). Chromatographic grade acetonitrile, n-hexane, common acid-base reagents (hydrochloric acid, sodium hydroxide), and amino acid analysis-specific reagents (sulfosalicylic acid, components of phosphate buffer) were all purchased from Aladdin Biochemical Technology Co., Ltd. (Shanghai, China). L-phenylalanine standard was obtained from Solarbio Science and Technology Co., Ltd. (Beijing, China). D (+)-anhydrous glucose, D-fructose, and sucrose were purchased from Yuanye Biotechnology Co., Ltd. (Shanghai, China). Anhydrous ethanol was purchased from Xilong Scientific Co., Ltd. (Guangzhou, China). The experimental water used was distilled water from Watsons (Guangzhou, China).

### 2.2. Sample Preparation

200 g of camellia seeds were roasted at 170 °C for 0, 5, 10, 15, 20, 25, and 30 min in a convection oven (Midea PT3520W, Foshan, China). A part of the roasted camellia seeds was crushed using an ultrafine pulverizer (SS-1022, Wuyi Haina Electric Appliance Co., Ltd., Jinhua, China) and then passed through a 60-mesh sieve for subsequent analysis. The remaining camellia seeds were pressed using the cold pressing method by a small screw press machine (DH-58, Daohang Machinery Co., Ltd., Dezhou, China) to obtain the crude oil, which was centrifuged at 2000× *g* for 15 min in a high-speed centrifuge (TDL-40B, Shanghai Anting Scientific Instrument Factory, Shanghai, China) at room temperature. The upper clear oil fraction was then degummed and vacuum-dried All samples were stored at 4 °C before analysis.

### 2.3. Free Amino Acid Analysis

The analysis of free amino acids was performed based on the method described in a previous publication [22] with a slight modification. Roasted camellia seed powder (1 g) was extracted with 50 mL of 0.01 mol/L hydrochloric acid for 30 min under constant stirring. After filtration, 2 mL of the filtrate was added to 2 mL of 8% (w/v) sulfosalicylic acid, and the mixture was allowed to stand for a specified duration. Subsequently, the mixture was centrifuged at 5000× *g* for 10 min at room temperature, after which the supernatant was filtered through a 0.22 µm membrane filter (Shimadzu-GL WondaDisc MCE, Kyoto, Japan). Using ninhydrin as the derivatizing reagent and in combination with post-column derivatization method, the samples were then analyzed for amino acids using an amino acid analyzer (SYKAM S-433D, Sykam GmbH, Munich, Germany) equipped with an ultraviolet spectrophotometer detector and PEEK columns (4.6 mm × 150 mm, SYKAM LCA K06/Na, Sykam GmbH, Munich, Germany). The injection volume was 50 μL, and the column oven temperature was maintained at 37 °C. The mobile phase consisted of A (0.04 M trisodium citrate dihydrate and 0.03 M citric acid, pH 3.5), B (0.07 M trisodium citrate dihydrate and 0.08 M boric acid, pH 10) and C (0.2 mg/mL ethylenediaminetetraacetic acid) using a gradient elution. The flow rate of the elution pump was 0.45 mL/min, and that of the derivatizing solution was 0.2 mL/min. The limit of detection (LOD) was 0.002 mg/g defatted camellia seeds. External standard quantification was employed. Standard curves were established for each of the 17 amino acids in the mixed standard mixture. The measured amino acid peak areas were input into the standard curve equations for quantitative calculation.

### 2.4. Determination of Soluble Sugars

The determination of soluble sugar was performed using high-performance liquid chromatography (HPLC) according to the previous research [23] with a slight modification. Referring to GB 5009.6-2016 [24], fats were extracted from camellia seed powder using the Soxhlet extraction method. The defatted residue was air-dried at room temperature to volatilize residual solvents, then dried at 45 °C until constant weight. Then, the defatted camellia seed powder was suspended in 10 mL of 70% ethanol and subjected to ultrasonic-assisted extraction, followed by centrifugation at 1000× *g* for 10 min at room temperature. The extraction process was performed twice. Then the combined supernatant was filtered and concentrated to 2 mL. Subsequently, the concentrated extract was centrifuged at 5000× *g* for 10 min at room temperature, and the resulting supernatant was collected for subsequent analysis. The analysis was conducted on an HPLC (Agilent 1260, Agilent Technologies Inc., Santa Clara, CA, USA) equipped with a refractive index detector (RID) and a Spherisorb NH_2_ column (4.6 × 250 mm, 5 µm). The column temperature was set at 30 °C and the injection volume was 5 μL. The mobile phase was 70% acetonitrile, delivered at a flow rate of 1.0 mL/min. The limit of detection (LOD) was 0.01 mg/g. Quantitative calculations were performed using the external standard method. Standard curves for the reference materials were constructed based on the peak area and concentration, and then the sugars contents were calculated according to these curves.

### 2.5. Volatile Component Analysis

Following the method of Yin et al. [25] with slight modifications, headspace solid-phase microextraction coupled with gas chromatography-mass spectrometry (HS-SPME/GC-MS) was used for the analysis of volatile components in CO. 10 g of CO was transferred to a 15 mL headspace vial and equilibrated at 60 °C for 20 min. And 10 μL of 1,2-o-dichlorobenzene (18.82 mg/kg) was also added to the sample vial as an internal standard. The HS-SPME fiber with a DVB/PDMS/CAR (2 cm, 50/30 μm, SUPELCO, Merck KGaA, Darmstadt, Germany) composite coating was vertically inserted into the headspace vial to perform headspace adsorption at 60 °C for 30 min. The adsorbed fiber was transferred to the injection port of GC-MS (Agilent 7890A, Agilent Technologies Inc., Santa Clara, CA, USA) and desorbed at 250 °C for 3 min to complete sample enrichment.

The analytical parameters of the GC-MS system were as follows: HP-5MS capillary column (30 m × 0.25 mm × 0.25 μm) was used. The oven temperature was initially maintained at 30 °C for 5 min, then increased to 100 °C at a rate of 6 °C/min and held for 5 min, and then increased to 180 °C at the same rate and maintained for 3 min. Helium (purity ≥ 99.999%) was used as the carrier gas at a flow rate of 0.8 mL/min, and the inlet temperature was set to 250 °C. The ionization source was operated in EI mode at an ionization energy of 70 eV. The temperatures of the ion source and transmission line were set at 230 °C and 280 °C, respectively, and the quadrupole temperature was controlled at 150 °C. The mass scanning range was set at 28–500 u. Compound identification was performed by comparing the obtained mass spectra with those in the National Institute of Standards and Technology (NIST) mass spectral library. Relative contents of all components were calculated using the area normalization method.

### 2.6. Preparation of Reaction Model Systems

According to the method of Hu et al. [26] with minor adjustments, the reaction model systems were constructed. The systems are as follows:

Maillard reaction model system: Amino acids (Arginine, Glutamic acid, Glycine, Histidine, Leucine, Phenylalanine, and Lysine) and glucose, each at a concentration of 0.15 mol/L, were mixed in 18 mL of phosphate buffer (0.2 mol/L, pH 7.0), and heated at 150 °C for 2 h.

Lipid oxidation model system: 6 g of unroasted cold-pressed CO was mixed with 18 mL of phosphate buffer (0.2 mol/L, pH 7.0), and then heated at 150 °C for 2 h.

Maillard-lipid oxidation interaction model system: 6 g of unroasted cold-pressed CO was added to 18 mL of the Lys–glucose Maillard model system, and then heated at 170 °C for 2 h.

The volatile flavor compounds in model systems were analyzed by HS-SPME-GC/MS, with unroasted cold-pressed CO used as the blank control. Each sample from different systems (4 g) was mixed with 10 μL of 1,2-dichlorobenzene internal standard solution (4.0 mg/kg) in a 20 mL headspace vials for processing and subsequent detection. All other conditions were consistent with Section 2.5.

### 2.7. Statistical Analysis

All treatments were performed in triplicate. The Partial Least Squares Regression (PLSR) model was established using Unscrambler 9.0. IBM SPSS Statistics 26.0 was used for one-way analysis of variance (ANOVA) and Duncan test to compare the differences among means. A level of probability of *p* < 0.05 was set as statistically significant. Graphs were plotted using Origin 2025.

## 3. Results and Discussion

### 3.1. Changes in Contents of Volatile Compounds During Roasting

The volatile components of CO with different roasting durations (0, 5, 10, 15, 20, 25, and 30 min) were analyzed using HS-SPME-GC/MS (Table 1). A total of 38 compounds were identified, including aldehydes, heterocycles, ketones, esters, alcohols, acids, and other substances. The results showed that roasting treatments significantly affected the volatile profile of CO. As roasting time increased, the contents of aldehydes, heterocycles, ketones, and acids increased, with the most pronounced changes observed in aldehydes, acids, and heterocycles. The formation of these compounds was mainly attributed to the Maillard reaction, followed by oxidative degradation [27], highlighting the critical role of roasting in developing the characteristic aroma of CO.

Aldehydes are generally regarded as the key volatile aroma components in CO [28]. The result showed that the types and concentrations varied dynamically during roasting. Their formation was associated with both the Maillard reaction and lipid oxidation [29]. In particular, heptanal, hexanal, nonanal, and isovaleraldehyde were detected in all samples. The concentrations of these aldehydes continued to increase during the early stage of roasting and were further promoted by thermal reactions in the middle stage, but decreased in the late stage (under 30 min roasting) due to excessive heating. This may be attributed to physical volatilization under sustained high temperatures. Additionally, the unsaturated aldehydes initially formed by lipid oxidation are highly reactive intermediates that may undergo various secondary reactions (e.g., oxidative decomposition and thermal degradation), leading to reduced concentrations [27,30]. However, unsaturated aldehydes such as (E)-2-heptenal and 2-phenyl-2-butenal were only formed during the middle and late stages. (E)-2-heptenal was primarily generated through the oxidative degradation of linoleic acid [31]. Although it typically exhibits a strong unpleasant fatty and grassy odor, it contributes to the overall aroma profile at extremely low concentrations. In contrast, the formation of 2-phenyl-2-butenal involves the interactions between the Maillard reaction and lipid oxidation, and it imparts a rich fruity, floral, and coumarin-like aroma that contributes to the unique flavor of CO [32,33].

No heterocycles were detected in the unroasted CO samples. With prolonged roasting, both the diversity and content of heterocycles increased significantly after 15 min, predominantly including pyrazines, pyrroles, and pyrans. These compounds were primarily generated through the Maillard reaction between free amino acids and reducing sugars, and could also form via the Strecker degradation reaction between α-amino acids and α-dicarbonyl compounds [34]. Most pyrazines exhibited a progressive increase during roasting, such as 2,5-dimethylpyrazine and 3-ethyl-3,5-dimethylpyrazine. These compounds have also been identified in other roasted oilseeds (e.g., peanuts, sesames, and almonds) [35], and contribute substantially to roasted and nutty aromas owing to their low sensory threshold [36]. Pyrroles and pyridines are N-heterocyclic compounds that co-occur with pyrazines during the Maillard reaction and are primarily responsible for imparting smoky and burnt aromas [37]. Furfural is primarily produced via sugar degradation in the Maillard reaction or caramelization, providing a caramel-like aroma [38].

In addition to aldehydes and heterocycles, the concentrations of acids, ketones, and esters also showed dynamic changes with increasing roasting time. The concentrations of acids were relatively low in the initial stage, and exhibited a fluctuating trend during subsequent heating due to the effect of lipid oxidative degradation reactions. Such compounds typically exhibit pungent and irritating aromas [39]. Ketones were mainly generated in the middle and late stages of roasting, with 2-decanone and 2-nonanone accounting for a relatively high proportion. They are usually formed via fatty acid autoxidation and carbohydrate thermal degradation, which mainly contribute caramel and fruity aromas in CO [40]. Among the esters identified, γ-butyrolactone and methyl glyoxylate had relatively high concentrations and typically imparted fruity and floral aromas to CO. Hydrocarbons such as hexane, 2,2,4-trimethylpentane, and (±)-limonene made minimal contributions to the oil’s aroma profile, primarily owing to their high sensory thresholds that restrict their involvement in flavor formation [7].

### 3.2. Changes in Contents of Free Amino Acid During Roasting

The formation mechanism of aroma compounds in camellia seeds during roasting primarily involves the Maillard reaction [41]. This reaction relies on specific precursors, with amino acids acting as key nitrogen sources and plays a crucial role in the reaction system by participating in a series of complex chemical reactions, including carbonyl-amine condensation, rearrangement, degradation, and polymerization with reducing sugars [42].

The changes in free amino acids in camellia seeds during roasting are shown in Table 2. A total of 17 free amino acids were identified. In unroasted defatted camellia seeds, arginine (Arg) had the highest content at 1.91 mg/g, followed by glutamic acid (Glu), lysine (Lys), and alanine (Ala) with contents ranging from 0.47 to 0.68 mg/g. After roasting, the content of each free amino acid decreased significantly, with the total amino acid content dropping from 5.85 mg/g to 3.40 mg/g. Notably, Lys showed the highest reduction rate (79.49 ± 1.54%), followed by Gly and Tyr. However, neither was considered a dominant amino acid in camellia seeds due to their low initial contents During roasting, these amino acids are primarily involved in reactions, including Maillard reaction, Strecker degradation, decarboxylation, and deamination [43], resulting in significant differences in the types and contents of the final volatile aromas, which endow the CO with a unique flavor. Furthermore, free amino acids were consumed to varying degrees throughout the roasting process. Based on changes in initial amino acid contents and their relative ratios, arginine, glutamic acid, glycine, histidine, leucine, phenylalanine, and lysine were identified as the primary amino acid precursors contributing to the distinctive aroma of roasted camellia seeds. Consequently, these seven amino acids were selected as substrates for subsequent Maillard reaction model systems.

The free amino acids of camellia seeds not only serve as direct precursors for flavor formation during roasting but also highlight the nutritional value of this resource. The chemical nature of amino acid side-chain groups directly determines their reactivity during roasting and their contribution to CO’s quality. During roasting, the content of non-polar aliphatic amino acids with hydrophobic branched alkyl chains (e.g., glycine, leucine) decreased significantly. These amino acids readily interact with lipids and serve as key precursors for short-chain alkanes and aldehydes, contributing to the development of nutty flavor in CO [44]. Certain neutral amino acids (e.g., cysteine) possess -SH groups that readily oxidize to form disulfide bonds. The degradation produces compounds such as hydrogen sulfide, which enhances the roasted flavor profile. Additionally, phenylalanine, an aromatic amino acid containing a benzene ring, undergoes thermal degradation to form phenethyl alcohol, which imparts floral and fruity aromas. Some acidic amino acids (e.g., glutamic acid) and basic amino acids (e.g., arginine) possess charged carboxyl for acidic and guanidine for basic, respectively, rendering them highly soluble in the aqueous phase of camellia seeds and reactive in the Maillard reaction. These properties all contribute to the formation of flavor characteristics [45]. From a human nutritional perspective, roasting also exerts distinct effects on the retention of different types of amino acids. The results indicated that both essential and non-essential amino acid levels in camellia seeds decreased significantly after roasting, with the total reduction in essential amino acids being more pronounced. These variations revealed an inherent link between the nutritional components and their flavor potential in camellia seeds, confirming that the free amino acids act as both nutritional carriers and flavor precursors during roasting.

### 3.3. Correlation Analysis of Volatile Components and Free Amino Acids

Based on partial least squares regression (PLSR) analysis, this study investigated the correlation between free amino acids and volatile aroma compounds in camellia seeds across different roasting times. The constructed PLSR model comprised two principal components, PC1 and PC2, with the cross-validated explained variance reaching 93%, indicating that the model possessed strong explanatory power.

The correlation analysis results are shown in Figure 1. Except for the seven amino acids (Cys, Met, Ala, Val, Gly, Ile, and Pro), other amino acids fell within the 50–100% explained variance range, indicating a good model fit. Amino acids were predominantly distributed on the right side of the loading plot, while volatile substances, including aldehydes, heterocycles, and acids, were concentrated on the left side. Furthermore, these two groups of substances were located in different quadrants, indicating a complex correlation between free amino acid content and the formation of characteristic aroma compounds. Additionally, volatile substances such as γ-butyrolactone, 3-methyl-2-butenyl benzoate, methyl glyoxylate, valeric acid, and hexane exhibited significantly stronger correlations with free amino acids. These associations stem from complex amino acid reactions during roasting, including Strecker degradation, the Maillard reaction, and catalytic regulation. Specifically, γ-butyrolactone was mainly synthesized from the Maillard reaction products of glutamic acid through hydroxy acid lactonization, while valeric acid was mainly produced through Strecker degradation of leucine and lipid oxidation involving methionine. These results further indicate amino acids may directly provide precursors for the synthesis of volatile substances or regulate that lipid oxidation and esterification reactions through intermediates, thereby participating in the formation of the characteristic flavor components of CO.

### 3.4. Changes in Contents of Soluble Sugar During Roasting

During roasting, endogenous soluble sugars in camellia seeds (e.g., glucose, fructose, sucrose, pentoses) served as key precursors in a series of chemical reactions, including the Maillard reaction and caramelization, which form the basis for the distinctive color and flavor profile of roasted CO [46]. The composition and contents of soluble sugars in camellia seeds during roasting are shown in Table 3. The soluble sugars were predominantly composed of sucrose and glucose, with minor amounts of fructose. Fructose was only detected in unroasted camellia seed and at relatively low levels, but was not detected after roasting. Although sucrose did not directly contribute to flavor formation, its content decreased significantly with prolonged roasting, likely due to the hydrolysis of sucrose into glucose or fructose during roasting [47]. Furthermore, high temperatures could induce sucrose caramelization, leading to dehydration and the formation of caramel-like compounds, which imparted complex sweet notes to the aroma of roasted CO. Additionally, glucose content varied inconsistently during roasting, and did not increase proportionally to the extent of sucrose degradation, indicating that glucose was also consumed during roasting. As a reducing sugar, glucose not only underwent caramelization during roasting but also participated in the Maillard reactions with amino acids, generating numerous volatile compounds (e.g., furans, pyrazines, aldehydes). These compounds are the core substances that constitute the nutty, burnt and roasted aromas of roasted CO. Therefore, glucose and sucrose are preliminarily inferred to be important precursor substances for the formation of characteristic aromas in roasted CO.

### 3.5. Correlation Analysis of Volatile Components and Soluble Sugars

The transformation dynamics between soluble sugars and volatile aroma compounds during the thermal processing of camellia seeds were systematically analyzed using a PLSR model (Figure 2). The cumulative explained variance of the two principal components, PC1 and PC2, accounted for 82%, indicating that the model exhibited strong predictive capability. Sucrose, as a key precursor substance, was distributed within the 50–100% explained variance interval and on the right side of the loading plot. In contrast, the volatile components such as aldehydes, heterocycles, ketones, and acids clustered on the left side of the loading plot, revealing a negative correlation between sucrose and the volatile constituents. Sucrose was mainly degraded into glucose and fructose during thermal processing. Glucose exhibited a stronger significant correlation with characteristic aroma compounds and showed negatively correlated with most volatile compounds, mainly including heterocycles (such as pyrazines, pyrroles), some Strecker-degraded aldehydes (e.g., nonanal, benzaldehyde), and pyranone compounds. Their formation all relied on glucose as a reducing sugar to participate in the Maillard reaction, ultimately forming volatile products with roasted, nutty, and caramel aromas. This finding not only confirms the previous inferences but also provides new experimental evidence for elucidating the formation mechanisms of the characteristic aroma in roasted CO. Based on the result, glucose was selected as the main precursor substance for the study of the simulation systems, in order to focus on the core monosaccharide stage of the Maillard reaction.

### 3.6. Flavor Profile Analysis in the Reaction Model Systems

Figure 3 shows the relative contents of volatile flavor compounds generated from different Maillard reaction model systems, including reactions of Arg, Glu, Gly, His, Leu, Phe, and Lys with glucose, the lipid oxidation reactions and the interaction system between “lys–glucose” and lipid oxidation. As indicated in Figure 3, the volatile components formed in each model system mainly included heterocycles, aldehydes, acids, esters, alcohols, and ketones. Heterocycles were primarily originated from the Maillard reaction and its synergistic interaction with lipid oxidation, exhibiting the highest relative abundance in the Lys–glucose reaction system. In contrast, no heterocycles were detected in the sole lipid oxidation model system. This is because simple lipid oxidation produces only linear or slightly branched carbonyl compounds, which lack heterocyclic structures. However, lipid oxidation can provide important precursors for the formation of heterocycles [25]. Aldehydes were generated in all model systems except the His-based system, although their content distribution exhibited significant variations. Esters accounted for a significant proportion in the lipid oxidation system, which may be attributed to the esterification reaction during the lipid oxidative degradation. Conversely, the relative contents of alcohols and acids were low in all systems, indicating their limited production. This may be attributed to the instability of these compounds in the high-temperature dynamic system, where they likely serve as reaction intermediates and are converted into more stable products, including alkenes, alkanes, and aromatic compounds such as aldehydes, ketones, esters, and heterocycles [30].

### 3.7. Formation of Aroma Compounds Under Simulated Reaction Systems

The profiles of volatile aroma compounds produced in the simulated reaction systems are shown in Table 4. A total of 47 volatile compounds were identified across seven Maillard reaction model systems, including 2 alcohols, 6 aldehydes, 1 acid, 8 ketones, 3 alkenes, 6 esters, and 21 heterocycles. The lysine–glucose model system produced the greatest diversity of heterocycles, confirming that these compounds were primarily generated via the Maillard reaction. Moreover, the results indicated that the dominant components of the Maillard reaction were regulated by amino acid structure. Basic amino acids (e.g., Lys) tended to produce pyrazine-type volatile compounds like 2,5-dimethylpyrazine, yielding baked aromas; aromatic amino acids (e.g., Phe) readily produced benzene-ring compounds like styrene and acetophenone; branched-chain amino acids (e.g., Leu) yielded pyridine derivatives (e.g., 2-hexanoylpyridoxine) as characteristic products, while neutral and acidic amino acids (e.g., Arg, Glu) yielded volatile compounds at lower concentrations with milder flavor contributions.

A total of 31 volatile compounds were generated through the lipid oxidation model system, including 1 alcohol, 13 aldehydes, 3 acids, 6 ketones, 1 alkene, 4 esters, and 3 heterocycles. In the lipid oxidation model system, both the variety and content of aldehydes increased significantly, with values exceeding those observed in the Maillard reaction model systems, indicating that aldehydes were mainly derived from the oxidative degradation of oil during thermal processing. Among them, benzaldehyde and phenylacetaldehyde, as representatives of the aromatic group, are considered to be mainly the products of the Strecker degradation reaction of phenylalanine precursors. These aldehydes exhibited pleasant odors. Additionally, this system generated furan-type heterocycles, while no nitrogen-containing heterocycles (e.g., pyrazines, pyrroles) were detected. This is primarily because the system operates within a chemical environment dominated by lipid oxidation, and contains only trace amounts of nitrogen sources. Furan is a typical product of lipid oxidation, directly derived from the thermal degradation of unsaturated fatty acids. The formation of pyrazine and pyrrole compounds depend on the Maillard reaction. The trace nitrogen sources in the system were insufficient to drive the formation of complex nitrogen-containing heterocycles, which required high nitrogen concentrations and specific pH conditions, except for some simple nitrogen-containing molecules (e.g., methyl N-hydroxybenzimidate, which was detected). It indicated that amino acids are the key nitrogen sources of nitrogen-containing flavor heterocycles in roasted camellia seeds.

In the simulated system combining the lysine–glucose Maillard reaction with lipid oxidation, heterocycles, ketones, aldehydes, and acids were the predominant reaction products. Upon adding unroasted cold-pressed CO, the compounds such as 5-methyl-2-furanmethanol, trans-2,4-decadienal, hexanoic acid, and 1-(3,4,5,6-tetrahydropyridin-2-yl) ethanone were predominantly generated in the simulated system, likely attributable to interactions between abundant aldehydes in CO and lysine, imparting a rich and intense roasted aroma to the CO [48]. In the cross-reaction system, 1,5-dihydro-3,4-dimethyl-2H-pyrrol-2-one was mainly produced via alkylation reactions, enhancing the caramel and nutty flavors of the systems. 2-ethyl-3,5-dimethylpyridine primarily formed through carbonyl-amine condensation and cyclization reactions in the cross-reaction system and served as a key contributor to the aromas of nuts, cocoa, coffee, and toasted bread. 2,4,4-trimethyl-3-(3-methylbutyl)-2-cyclohexen-1-one was predominantly generated via aldehyde-aldehyde condensation and cyclization reactions, and these complex cyclic ketone compounds impart a richer smoky and woody character to CO, increasing its thickness and persistence. Pyrazines were mainly produced by the Maillard reaction between nitrogen-containing precursors in CO (including free amino acids, peptides and proteins) and reducing sugars, with the Strecker degradation reaction occurring concurrently during roasting. Considering that the simulated system altered the internal microenvironment of the seeds, thereby affecting the buffering and protective functions of the internal matrix, a reaction temperature was selected to enable more precise sampling and tracking of key volatile flavor compounds across different systems. The experiments demonstrated that the core reaction pathways remained consistent at both 170 °C and 150 °C. Therefore, this study confirms that the Maillard reaction, lipid oxidation, and their cross-interactions involve a series of complex chemical reactions that can alter the content and composition of aroma compounds compared to those generated in the individual reactions, thereby contributing to the formation of flavor compounds with more distinctive characteristics.

In addition, by comparing the volatile compounds generated in the simulated systems (Table 4) with those produced during the roasting of CO at different roasting times (Table 1), a total of 18 overlapping volatile compounds were identified. Among these, aldehydes (e.g., (E)-2-heptenal, heptanal, octanal, hexanal), straight-chain ketones (e.g., 2-nonanone, 2-nonanal), and acids (e.g., heptanoic acid, octanoic acid) were primarily derived from lipid oxidation reactions. Heterocycles (e.g., 2,3,5-trimethylpyrazine, 2,5-dimethylpyrazine) were primarily generated through the Maillard reaction, particularly from the interactions between leucine or lysine and glucose, and their concentrations gradually increased during roasting. The synergistic interaction between the Maillard reaction and lipid oxidation enhanced the yield of key aromatic compounds (e.g., 2,5-dimethylpyrazine) and generated synergistic products (e.g., 2-ethyl-2,5-dimethylpyridine). The results indicated that the Maillard reaction primarily conferred the characteristic roasted, nutty, and caramel-like flavor notes in roasted CO, whereas lipid oxidation predominantly generated green, fresh, or fatty sensory attributes. The synergistic interactions between these two core metabolic pathways not only reinforced the integrated flavor profile but also underscored the pivotal role of cross-regulation between them in modulating the complex composition of aroma-active compounds in CO.

In summary, these liquid simulation systems served as simplified models for the low-moisture solid matrix of roasted camellia seeds, allowing for precise isolation of precursor-product relationships by eliminating interference from the solid matrix and enabling investigation into the fundamental mechanisms underlying flavor compound formation. Comparing the results from both matrices revealed that the liquid matrix generated key volatile compounds similar to those present in the solid matrix, including aldehydes and heterocycles. This indicated that although the liquid matrix differs fundamentally from the seed solid matrix in terms of water activity, concentration of precursor compounds, mass transfer and thermodynamic properties, the basic chemical pathways of the Maillard reaction, lipid oxidation and their interactions revealed by the simulation systems undoubtedly constitute the core mechanism driving the actual flavor formation of CO.

## 4. Conclusions

In conclusion, this study combined PLSR models to confirm that free amino acids and soluble sugars exhibited strong correlations with the volatile flavor compounds in CO at different roasting durations. Arg, Glu, Gly, His, Leu, Phe, and Lys, as well as sucrose and glucose, are regarded as the primary precursors for the characteristic aromas of roasted CO. 38 volatile compounds were identified in the roasting seeds oil for 0–30 min, while 80 compounds were detected in the simulated reaction systems, revealing the formation pathways of flavor compounds. Specifically, aldehydes, acids, and ketones were mainly derived from the thermal oxidative degradation of lipids. Esters originated from the different simulated Maillard reaction systems. And heterocycles were primarily generated in the Maillard simulated systems and the co-reaction system of the Maillard reaction and lipid oxidation. Specially, the “Lys–glucose” system produced the largest variety and highest concentration of heterocycles. Therefore, significant interactive effects of the Maillard reaction and lipid oxidation on the aroma formation of CO were observed in the model reaction systems, which enriched its unique flavor profile. These findings contribute to understanding the potential formation pathways of flavor compounds in CO. Future research may incorporate stable isotope tracing technology to further elucidate the formation pathways of key aromatic compounds. Furthermore, it is recommended to construct reaction kinetics models under similar or even identical temperature gradients, and to develop model systems with controlled water activity or reaction systems on solid substrates, in order to more accurately and quantitatively describe the dynamic change patterns of key aroma components under different conditions. The molecular mechanisms underlying the formation of characteristic aroma in CO could be further clarified by obtaining key parameters such as reaction rate constants, ultimately achieving controllable regulation of its flavor quality.

## Figures and Tables

**Figure 1 foods-15-00087-f001:**
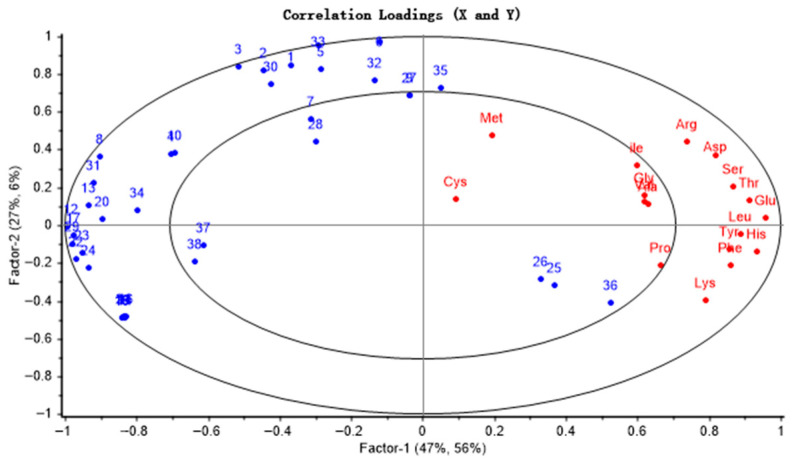
Factor loadings of PLS components 1 and 2 on 38 GC-MS peaks and free amino acid from camellia seeds. (1 to 38 correspond to the numbers of volatile substances in Table 1).

**Figure 2 foods-15-00087-f002:**
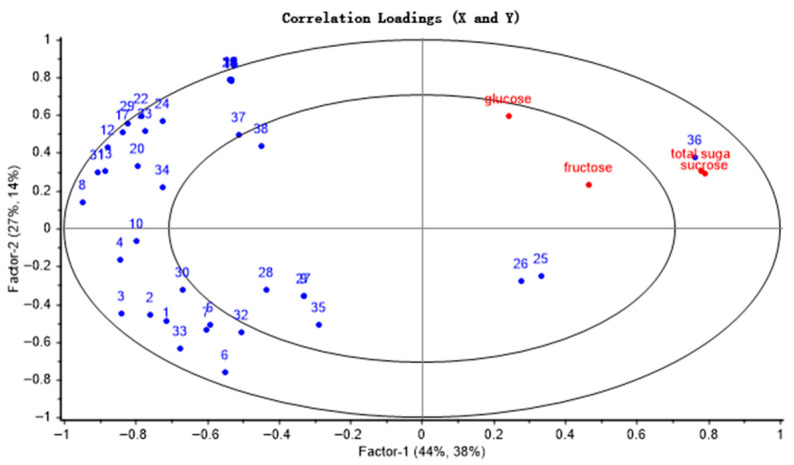
Factor loadings of PLS components 1 and 2 on 38 GC-MS peaks and soluble sugar from camellia seeds. (1 to 38 correspond to the numbers of volatile substances in Table 1).

**Figure 3 foods-15-00087-f003:**
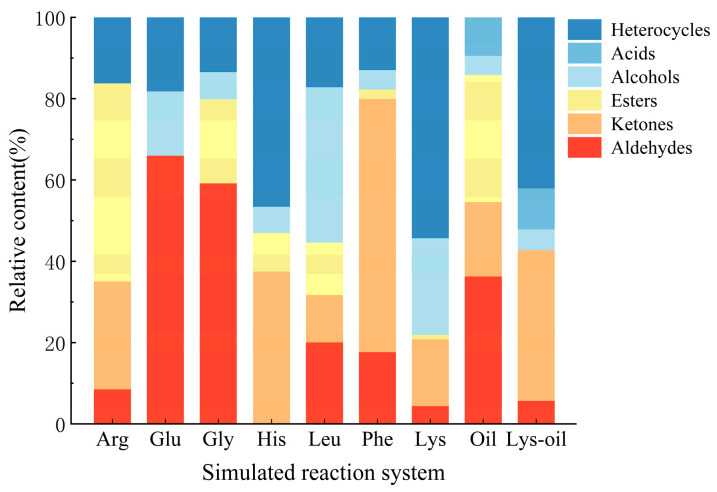
Histogram of relative content of flavor substances in different simulated reaction systems.

**Table 1 foods-15-00087-t001:** Volatile compounds identified in the head-space of camellia seeds during different roasting time using HS-SPME-GC/MS.

Number	Compounds	RI	Relative Content (mg/kg)						
			0 min	5 min	10 min	15 min	20 min	25 min	30 min
Aldehydes									
1	Heptanal	901.22	0.09 ± 0.02 ^e^	0.10 ± 0.02 ^e^	0.16 ± 0.01 ^d^	0.17 ± 0.01 ^d^	0.72 ± 0.02 ^a^	0.62 ± 0.02 ^b^	0.27 ± 0.02 ^c^
2	Hexanal	794.80	0.42 ± 0.03 ^f^	0.20 ± 0.02 ^g^	0.66 ± 0.03 ^d^	0.59 ± 0.02 ^e^	1.99 ± 0.03 ^a^	1.78 ± 0.02 ^b^	0.85 ± 0.02 ^c^
3	Nonanal	1105.63	0.27 ± 0.03 ^e^	0.09 ± 0.01 ^f^	0.37 ± 0.03 ^d^	0.69 ± 0.03 ^c^	1.34 ± 0.06 ^a^	1.16 ± 0.09 ^b^	0.68 ± 0.03 ^c^
4	Pentanal	967.27	0.05 ± 0.02 ^e^	-	0.26 ± 0.01 ^d^	0.74 ± 0.03 ^ab^	0.43 ± 0.03 ^c^	0.69 ± 0.03 ^b^	0.77 ± 0.03 ^a^
5	Decanal	1207.25	-	-	-	-	0.05 ± 0.02	0.06 ± 0.02	-
6	Octanal	1003.58	-	0.13 ± 0.01 ^d^	0.44 ± 0.03 ^c^	1.14 ± 0.06 ^b^	2.07 ± 0.05 ^a^	2.10 ± 0.07 ^a^	-
7	Benzaldehyde	967.27	-		0.32 ± 0.03 ^c^	0.80 ± 0.02 ^ab^	0.47 ± 0.03 ^b^	0.50 ± 0.04 ^b^	0.32 ± 0.02 ^c^
8	Isovaleraldehyde	675.29	0.13 ± 0.02 ^d^	0.20 ± 0.03 ^d^	0.20 ± 0.01 ^d^	0.61 ± 0.04 ^c^	2.76 ± 0.05 ^b^	3.66 ± 0.10 ^a^	3.57 ± 0.09 ^a^
9	(E)-2-Heptenal	958.02	-	-	-	-	0.08 ± 0.01	-	-
10	2-Phenyl-2-butenal	1281.03	-	-	-	0.09 ± 0.02 ^a^	0.04 ± 0.01 ^b^	0.08 ± 0.01 ^a^	0.08 ± 0.02 ^a^
Heterocycles									
11	2-Methylpyrazine	821.59	-	-	-	-	-	-	0.30 ± 0.02
12	2,5-Dimethylpyrazine	915.07	-	-	-	1.23 ± 0.03 ^c^	1.25 ± 0.05 ^c^	2.69 ± 0.03 ^b^	4.86 ± 0.09 ^a^
13	3-Ethyl-2,5-dimethylpyrazine	1086.70	-	-	-	0.34 ± 0.03 ^c^	0.26 ± 0.02 ^d^	0.45 ± 0.04 ^b^	0.67 ± 0.04 ^a^
14	2-Ethyl-3,5-dimethylpyrazine	1080.76	-	-	-	-	-	-	0.36 ± 0.02
15	2-Ethyl-6-methylpyrazine	1009.41	-	-	-	-	-	-	1.15 ± 0.10
16	3,5-Diethyl-2-methylpyrazine	1161.86	-	-	-	-	-	-	0.98 ± 0.04
17	2-Acetylpyrrole	1061.16	-	-	-	-	0.13 ± 0.02 ^c^	0.23 ± 0.02 ^b^	0.51 ± 0.03 ^a^
18	2-Acetyl-1,4,5,6-tetrahydropyridine	1199.41	-	-	-	-	-	-	0.66 ± 0.03
19	2-(Propan-2-yloxy)tetrahydro-2H-pyran	1399.08	-	-	-	-	-	-	0.96 ± 0.04
20	Furfural	832.90	-	-	-	-	0.06 ± 0.01 ^c^	0.39 ± 0.03 ^b^	0.49 ± 0.03 ^a^
Ketones									
21	2-Pyrrolidone	1067.46	-	-	-	-	-	-	0.72 ± 0.03
22	2-Decanone	1193.64	-	-	-	0.07 ± 0.02 ^d^	0.14 ± 0.02 ^c^	0.23 ± 0.04 ^b^	0.83 ± 0.03 ^a^
23	2-Nonanone	1092.07	-	-	-	0.02 ± 0.01 ^c^	0.04 ± 0.01 ^c^	0.28 ± 0.02 ^b^	0.50 ± 0.03 ^a^
24	3,5-Dihydroxy-6-methyl-2,3-dihydro-4H-pyran-4-one	1193.97	-	-	-	-	-	0.21 ± 0.02	0.60 ± 0.05
Esters									
25	γ-Butyrolactone	916.48	0.03 ± 0.01	-	0.24 ± 0.02	-	-	-	-
26	3-Methyl-2-butenyl benzoate	1078.77	-	-	0.14 ± 0.02	-	-	-	-
27	Methyl glyoxalate	613.16	-	-	-	-	0.30 ± 0.01	-	-
Alcohols									
28	5-Methyl-5-hexen-2-ol	1193.78	-	-	-	-	-	0.07 ± 0.01	-
29	2-Ethylhex-2-enol	1124.02	-	-	-	-	0.08 ± 0.01 ^b^	0.09 ± 0.02 ^b^	0.25 ± 0.03 ^a^
Acids									
30	Hexanoic acid	973.90	0.30 ± 0.04 ^c^	0.04 ± 0.02 ^d^	-	-	1.00 ± 0.06 ^b^	1.29 ± 0.04 ^a^	0.32 ± 0.06 ^c^
31	Heptanoic acid	1071.94	-	-	-	-	0.30 ± 0.02 ^b^	0.26 ± 0.02 ^b^	0.53 ± 0.03 ^a^
32	Nonanoic acid	1264.15	0.20 ± 0.03 ^e^	-	0.48 ± 0.03 ^c^	0.56 ± 0.03 ^b^	1.41 ± 0.03 ^a^	0.59 ± 0.03 ^b^	0.38 ± 0.04 ^d^
33	Octanoic acid	1165.36	-	-	0.07 ± 0.02 ^d^	0.36 ± 0.04 ^c^	0.77 ± 0.03 ^a^	0.69 ± 0.03 ^b^	0.11 ± 0.02 ^d^
34	Acetic acid	602.95	-	-	-	-	-	0.58 ± 0.03	0.42 ± 0.03
35	Valeric acid	1068.52	-	-	-	0.12 ± 0.02	0.31 ± 0.03	-	-
Other types									
36	Hexane	609.76	0.04 ± 0.02	0.05 ± 0.02	-	-	-	-	-
37	2,2,4-Trimethylpentane	673.31	0.12 ± 0.02 ^d^	0.25 ± 0.03 ^c^	0.30 ± 0.03 ^c^	-	1.04 ± 0.05 ^b^	-	1.21 ± 0.04 ^a^
38	(±)-Limonene	1033.10	0.19 ± 0.02 ^c^	0.05 ± 0.01 ^d^	-	0.28 ± 0.03 ^b^	-	0.25 ± 0.02 ^b^	0.41 ± 0.03 ^a^

Notes: Data are shown as mean ± standard deviation (n = 3). Compound content is relative to the internal standard concentration; “-“ indicates that the substance is not detected. Different letters in the same row are recognized as the statistically significant differences between roasting times for the same compounds (*p* < 0.05).

**Table 2 foods-15-00087-t002:** Changes in free amino acid content in camellia seeds during different roasting time (mg/g defatted camellia seeds).

Amino Acid	Roasting Time (min)							Rates of Decrease (%)
	0	5	10	15	20	25	30	
Asp	0.44 ± 0.01 ^a^	0.40 ± 0.01 ^c^	0.44 ± 0.00 ^a^	0.45 ± 0.01 ^a^	0.42 ± 0.00 ^b^	0.35 ± 0.01 ^d^	0.28 ± 0.01 ^e^	37.72 ± 3.16 ^fgh^
Ser	0.30 ± 0.01 ^bc^	0.35 ± 0.01 ^a^	0.29 ± 0.01 ^c^	0.34 ± 0.01 ^a^	0.32 ± 0.00 ^b^	0.24 ± 0.00 ^d^	0.18 ± 0.01 ^e^	39.85 ± 4.51 ^efg^
Thr	0.22 ± 0.01 ^abc^	0.24 ± 0.01 ^a^	0.20 ± 0.01 ^c^	0.23 ± 0.01 ^ab^	0.21 ± 0.01 ^bc^	0.18 ± 0.00 ^d^	0.12 ± 0.01 ^e^	46.04 ± 5.35 ^def^
Glu	0.68 ± 0.01 ^c^	0.72 ± 0.01 ^b^	0.74 ± 0.01 ^a^	0.68 ± 0.01 ^c^	0.62 ± 0.01 ^d^	0.48 ± 0.00 ^e^	0.31 ± 0.01 ^f^	53.85 ± 1.46 ^cd^
Gly	0.09 ± 0.01 ^a^	0.09 ± 0.01 ^a^	0.04 ± 0.00 ^c^	0.07 ± 0.01 ^b^	0.07 ± 0.01 ^b^	0.04 ± 0.01 ^c^	0.03 ± 0.00 ^c^	66.23 ± 1.85 ^b^
Ala	0.47 ± 0.02 ^abc^	0.50 ± 0.01 ^ab^	0.50 ± 0.02 ^a^	0.51 ± 0.02 ^a^	0.47 ± 0.03 ^bc^	0.44 ± 0.01 ^c^	0.35 ± 0.01 ^d^	25.50 ± 2.71 ^i^
Cys	0.03 ± 0.00 ^a^	0.03 ± 0.01 ^a^	0.03 ± 0.01 ^a^	0.04 ± 0.01 ^a^	0.03 ± 0.01 ^a^	0.03 ± 0.01 ^a^	0.02 ± 0.01 ^a^	30.06 ± 2.84 ^hi^
Val	0.14 ± 0.01 ^ab^	0.15 ± 0.01 ^a^	0.14 ± 0.01 ^ab^	0.16 ± 0.01 ^a^	0.14 ± 0.01 ^ab^	0.13 ± 0.01 ^b^	0.10 ± 0.01 ^c^	30.81 ± 1.04 ^hi^
Met	0.02 ± 0.00 ^ab^	0.02 ± 0.00 ^ab^	0.02 ± 0.01 ^ab^	0.02 ± 0.00 ^ab^	0.02 ± 0.00 ^ab^	0.03 ± 0.01 ^a^	0.01 ± 0.00 ^b^	52.76 ± 2.35 ^cd^
Ile	0.08 ± 0.01 ^ab^	0.09 ± 0.01 ^a^	0.07 ± 0.01 ^ab^	0.08 ± 0.01 ^a^	0.08 ± 0.01 ^ab^	0.07 ± 0.01 ^c^	0.05 ± 0.00 ^c^	31.70 ± 5.46 ^ghi^
Leu	0.13 ± 0.00 ^b^	0.15 ± 0.01 ^a^	0.13 ± 0.01 ^b^	0.13 ± 0.00 ^b^	0.11 ± 0.01 ^c^	0.09 ± 0.00 ^d^	0.06 ± 0.01 ^e^	57.22 ± 2.53 ^c^
Tyr	0.15 ± 0.01 ^a^	0.15 ± 0.00 ^a^	0.14 ± 0.02 ^a^	0.15 ± 0.01 ^a^	0.11 ± 0.01 ^b^	0.09 ± 0.01 ^b^	0.06 ± 0.01 ^c^	60.17 ± 1.99 ^bc^
Phe	0.27 ± 0.01 ^c^	0.29 ± 0.00 ^b^	0.31 ± 0.01 ^a^	0.27 ± 0.01 ^c^	0.23 ± 0.01 ^d^	0.19 ± 0.01 ^e^	0.14 ± 0.01 ^f^	48.20 ± 2.79 ^de^
His	0.33 ± 0.00 ^b^	0.36 ± 0.01 ^a^	0.33 ± 0.01 ^b^	0.32 ± 0.01 ^b^	0.27 ± 0.01 ^c^	0.23 ± 0.01 ^d^	0.16 ± 0.01 ^e^	52.23 ± 3.17 ^cd^
Lys	0.47 ± 0.00 ^a^	0.39 ± 0.00 ^b^	0.27 ± 0.00 ^c^	0.20 ± 0.01 ^d^	0.17 ± 0.01 ^e^	0.15 ± 0.01 ^f^	0.10 ± 0.01 ^g^	79.49 ± 1.54 ^a^
Arg	1.91 ± 0.01 ^d^	2.40 ± 0.01 ^a^	1.90 ± 0.01 ^d^	2.39 ± 0.00 ^a^	2.17 ± 0.02 ^b^	1.95 ± 0.01 ^c^	1.37 ± 0.01 ^e^	28.05 ± 0.24 ^i^
Pro	0.11 ± 0.00 ^b^	0.12 ± 0.00 ^ab^	0.12 ± 0.01 ^ab^	0.12 ± 0.01 ^a^	0.09 ± 0.01 ^c^	0.09 ± 0.00 ^c^	0.06 ± 0.01 ^d^	45.86 ± 4.64 ^def^
Total	5.85 ± 0.02	6.46 ± 0.03	5.68 ± 0.02	6.15 ± 0.01	5.51 ± 0.04	4.77 ± 0.03	3.40 ± 0.03	41.88 ± 1.56

Notes: Data are shown as mean ± standard deviation (n = 3). Different letters in the same row are recognized as the statistically significant differences during different roasting time for the same amino acid (*p* < 0.05). Different letters in the last column are recognized as the statistically significant differences during roasting for the rates of decrease in amino acid content (*p* < 0.05).

**Table 3 foods-15-00087-t003:** Soluble sugar content and composition analysis of camellia seeds during different roasting time.

Roasting Time (min)	Soluble Sugar (mg/g)			
	Sucrose	Glucose	Fructose	Total Sugar
0	86.24 ± 1.51 ^a^	1.77 ± 0.23 ^a^	0.53 ± 0.05	88.55 ± 1.70 ^a^
5	69.33 ± 3.13 ^b^	0.83 ± 0.18 ^c^	ND	70.17 ± 3.24 ^b^
10	44.29 ± 2.04 ^c^	0.72 ± 0.03 ^cd^	ND	45.00 ± 2.02 ^c^
15	45.25 ± 2.72 ^c^	0.52 ± 0.05 ^de^	ND	45.77 ± 2.70 ^c^
20	42.92 ± 1.31 ^cd^	0.93 ± 0.06 ^c^	ND	43.86 ± 1.30 ^cd^
25	39.48 ± 1.95 ^d^	0.43 ± 0.03 ^e^	ND	39.91 ± 1.98 ^d^
30	39.29 ± 1.27 ^d^	1.22 ± 0.15 ^b^	ND	40.51 ± 1.32 ^d^

Notes: Data are shown as mean ± standard deviation (n = 3). Different letters in the same column are recognized as the statistically significant differences during different roasting time for the same soluble sugar (*p* < 0.05).

**Table 4 foods-15-00087-t004:** Volatile components generated in the different simulated reaction systems.

Number	Compound	Molecular Formula	Odor	Simulated Reaction System	Average Relative Concentration (mg/kg)
Alcohols					
1	3-Furanmethanol	C_5_H_6_O_2_	caramel, sweet	2/3/4/5/6/9	0.754 ± 0.59
2	5-Methyl-2-furanmethanol	C_6_H_8_O_2_	caramel, sweet	9	1.084 ± 0.24
3	Phenethyl alcohol	C_8_H_10_O	greasy, sweet, roast	6	0.299 ± 0.06
4	Octanol	C_8_H_18_O	waxy, green, orange	8	0.844 ± 0.04
Aldehydes					
5	(E)-2-Heptenal	C_7_H_12_O	sweet, apple	8	0.323 ± 0.10
6	(E)-2-Hexadecenal	C_16_H_30_O	green, woody, apple	8	1.100 ± 0.03
7	(E,E)-2,4-Nonadienal	C_9_H_14_O	fatty, chicken, fried potato	8	0.206 ± 0.01
8	2,4,6-Trimethylbenzaldehyde	C_10_H_12_O	nut, bitter almond	2	0.463 ± 0.07
9	2,4-Heptadienal	C_7_H_10_O	nut, fatty	8	1.974 ± 0.18
10	2,4-Decadienal	C_10_H_16_O	fatty, orange, greasy	8/9	0.118 ± 0.02
11	2,5-Dimethylbenzaldehyde	C_9_H_10_O	bitter almond, caramel	1/2/3/4/5/6/7	0.522 ± 0.51
12	2-Undecenal	C_11_H_20_O	waxy, soapy	8/9	0.408 ± 0.05
13	3,4-Dimethylbenzaldehyde	C_9_H_10_O	nut, bitter almond	7/9	0.164 ± 0.07
14	Benzaldehyde	C_7_H_6_O	nut, bitter almond	6	0.102 ± 0.00
15	Phenylacetaldehyde	C_8_H_8_O	green	6	0.342 ± 0.03
16	4-Oxo-ketoaldehyde	C_9_H_16_O_2_	-	8	1.046 ± 0.01
17	trans-2-Octenal	C_8_H_14_O	spicy, cucumber, green	8	0.166 ± 0.00
18	trans-2,4-Decadienal	C_10_H_16_O	fatty, chicken, fried potato	9	0.417 ± 0.14
19	trans-2-Decenal	C_10_H_18_O	fatty, earthy, mushroom	8/9	0.856 ± 0.09
20	Heptanal	C_7_H_14_O	oil, green	8	1.266 ± 0.03
21	Decanal	C_10_H_20_O	fatty, orange, melon	8	0.260 ± 0.01
22	Hexanal	C_6_H_12_O	green, woody, apple	8	0.151 ± 0.00
23	Nonanal	C_9_H_18_O	cucumber, coconut	1/2/3/5/6/8/9	0.500 ± 0.22
Acids					
24	Heptanoic acid	C_7_H_14_O_2_	-	8	0.170 ± 0.00
25	Hexanoic acid	C_6_H_12_O_2_	sweat	9	0.294 ± 0.02
26	Nonanoic acid	C_9_H_18_O_2_	green, fatty	4/8/9	0.863 ± 0.06
27	Octanoic acid	C_8_H_16_O_2_	rancid, soapy, cheese	8	0.568 ± 0.20
Ketones					
28	1-(3,4,5,6-Tetrahydropyridin-2-yl) ethanone	C_7_H_11_NO	roast	9	1.963 ± 0.08
29	1-(3,4-Dihydropyridin-1(2H)-yl) ethanone	C_7_H_11_NO	roasted peanut, burnt	7/9	1.625 ± 1.84
30	2,4,4-Trimethyl-3-(3-methylbutyl)-2-cyclohexen-1-one	C_14_H_24_O	-	9	0.356 ± 0.02
31	2,6-Dihydroxy-3-methylacetophenone	C_9_H_10_O_3_	-	5/7	1.396 ± 1.74
32	4-Phenyl-2-butanone	C_10_H_12_O	chemistry, fruit, green	6	0.173 ± 0.00
33	6-Methyl-3-heptanone	C_8_H_16_O	fruit	5	0.283 ± 0.00
34	Acetophenone	C_8_H_8_O	cherry, bitter	6	1.153 ± 0.02
35	2-Butyl-2-cyclopenten-1-one	C_9_H_14_O	chemistry	8	2.635 ± 0.02
36	2-Decanone	C_10_H_20_O	orange, lemon	8	0.192 ± 0.00
37	2-Nonanone	C_9_H_18_O	cheese, green, butter	8	1.176 ± 0.00
38	3,4-Dimethyl-1,2-cyclopentanedione	C_7_H_10_O_2_	roast, caramel	8/9	1.754 ± 0.76
39	3-Octen-2-one	C_8_H_14_O	-	8	0.370 ± 0.01
40	Methylcyclopentenolone	C_6_H_8_O_2_	-	7/9	0.140 ± 0.02
41	Hydroxyacetone	C_3_H_6_O_2_	chemistry	1/7	0.350 ± 0.11
42	Ethylcyclopentenolone	C_7_H_10_O_2_	-	8/9	0.440 ± 0.32
43	2-Acetoxyacetone	C_5_H_8_O_3_	-	7	0.968 ± 0.03
Alkenes					
44	(2E)-1-Ethoxy-4,4-dimethyl-2-pentene	C_9_H_18_O	-	8	2.790 ± 0.01
45	(3Z)-Hexadec-3-ene	C_16_H_32_	fruit	5	0.494 ± 0.12
46	Styrene	C_8_H_8_	gasoline	6	3.373 ± 0.19
47	Dimethylhexene	C_8_H_16_	-	5	0.192 ± 0.01
Esters					
48	2,2,4-Trimethyl-1,3-pentanediol diisobutyrate	C_16_H_30_O_4_	pineapple, sweet, fruit	1/7	0.511 ± 0.41
49	Ethyl 5-hydroxypentanoate	C_19_H_30_O_3_	fruit, sweet, pineapple	5	1.287 ± 0.02
50	Methyl N-hydroxybenzimidate	C_8_H_9_NO_2_	fruit, sweet	1/2/3/8	0.331 ± 0.34
51	γ-Decalactone	C_10_H_18_O_2_	cream, fruit, nut	8	2.327 ± 0.22
52	4-Methylphenyl isocyanate	C_8_H_7_NO	green, sweet, spicy	3/4/7	0.234 ± 0.04
53	Vinyl hexanoate	C_8_H_14_O_2_	pineapple, waxy, fruit	8	0.535 ± 0.07
54	Dibutyl phthalate	C_16_H_22_O_4_	fruit, banana, peach	1/4/6/7	0.603 ± 0.67
55	Diisobutyl phthalate	C_19_H_28_O_4_	fruit, banana, peach	1/3/6/8/9	0.448 ± 0.37
Heterocycles					
56	1,5-Dihydro-3,4-dimethyl-2H-pyrrol-2-one	C_6_H_9_NO	sweet, tea	9	1.054 ± 0.00
57	2,3,4,5-Tetrahydropyridine	C_5_H_9_N	-	7	2.285 ± 0.03
58	2,3,5-Trimethylpyrazine	C_7_H_10_N_2_	cocoa, potato, barbecue	7/9	0.781 ± 0.15
59	2,3-Dimethyl-5-ethylpyrazine	C_8_H_12_N_2_	caramel, cocoa, roast	7/9	0.474 ± 0.17
60	2-Isoamyl-6- methylpyrazine	C_10_H_16_N_2_	-	9	0.141 ± 0.10
61	2,5-Dimethyl-4-hydroxy-3 (2H)-furanone	C_6_H_8_O_3_	caramel, marshmallow, honey	3/4/5/7/8	0.482 ± 0.36
62	2,5-Dimethylpyrazine	C_6_H_8_N_2_	roasted potato	7/9	1.267 ± 0.07
63	2-Hexanoylpyridine	C_11_H_15_NO	tobacco, rubber	5	2.617 ± 0.02
64	2-Methyl-3-propylpyrazine	C_8_H_12_N_2_	-	7/9	0.731 ± 0.05
65	2-Methyl-6-(3-methylbutyl) pyrazine	C_10_H_16_N_2_	-	5/7/9	0.639 ± 0.28
66	2-Methylpyrazine	C_5_H_6_N_2_	nut, scorch, roast	7/9	0.741 ± 0.22
67	2-Ethyl-3,5-dimethylpyridine	C_9_H_13_N	-	9	0.262 ± 0.01
68	2-Ethyl-5-methylpyrazine	C_7_H_10_N_2_	coffee, bean, eatage	7/9	0.317 ± 0.16
69	2-Ethyl-6-methyl-3-hydroxypyridine	C_8_H_11_NO	roast	9	0.235 ± 0.01
70	2-Ethyl-6-methylpyrazine	C_7_H_10_N_2_	roasted potato	3/7/9	0.474 ± 0.33
71	2-Acetyl-3-methylpyrazine	C_7_H_8_N_2_O	-	7/9	0.506 ± 0.01
72	2-Acetylpyrrole	C_6_H_7_NO	nut, walnut, roasted bread	3/4/5	0.341 ± 0.13
73	3,5-Dihydroxy-6-methyl-2,3-dihydro-4H-pyran-4-one	C_6_H_8_O_4_	-	1/3/4/5/6/7	0.596 ± 0.52
74	3-Methylpyrazine	C_5_H_6_N_2_	-	4/7/9	1.045 ± 0.42
75	3-Ethyl-2,5-dimethylpyrazine	C_8_H_12_N_2_	potato, roasted nut	7/9	0.193 ± 0.11
76	4-Methyl-2-(2-methylprop-1-enyl)pyridine	C_10_H_13_N	-	7/9	0.746 ± 0.05
77	4-Methyl-3,6-dihydroxypyrazine	C_5_H_6_N_2_O_2_	-	7	0.116 ± 0.00
78	Pyrazine	C_4_H_4_N_2_	-	7/9	0.916 ± 0.07
79	Furanone	C_6_H_8_O_3_	caramel	7/8/9	2.252 ± 0.15
80	Furfural	C_5_H_6_O_2_	woody, roasted bread, nut	5/8	0.499 ± 0.51

Notes: Different simulated reaction systems: Systems 1–7 represent Maillard simulated reaction systems of Arg, Glu, Gly, His, Leu, Phe, Lys with glucose, respectively; System 8 is a lipid oxidation simulated reaction system; System 9 is a simulated co-reaction system of “Lys–glucose” and lipid oxidation. The compound concentrations represent relative internal standard concentrations.

## Data Availability

The original contributions presented in the study are included in the article, further inquiries can be directed to the corresponding author.

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
