# Peer review of "Formation of Aroma Characteristics in Roasted Camellia oleifera Seeds"

_foods, 2025, doi:10.3390/foods15010087_

Round 1

Reviewer 1 Report

Comments and Suggestions for Authors

I have reviewed the manuscript titled “Formation of characteristic aroma in roasted Camellia oleifera seeds”. This paper assesses the effects of roasting (170 °C, 0-30 min) on free amino acids, soluble sugars, and volatile components in camellia seeds and the corresponding oils. The subject is interesting and presents some novelty. The paper is well presented and easy to read. The literature cited is relevant to the study. However, there are some weak points in the research, concerning overall design of the experiment and analytical chemistry. I think the paper could prove to be very interesting and useful to very large researchers, possibly making it acceptable for publication in Foods after major revisions.

I would like to give further suggestion on the following matters:

Line 117: Add the obtaining year of the camellia seeds

Line 119: How the authors were determined the maturity of the seeds?

Line 132: According to what criteria are the roasting conditions (time and temperarute) determined? It has to be explained.

Line 175: No clear data are reported concerning about the conditions used for SPME analysis, concerning time of equilibrium and its choice: were different trails carry out? Were trials finalised to assess time of equilibrium and analytes take off carried out? What about the linearity and repeatability applied analytical technique?

Line 176: Add the lenght of DVB/CAR/PDMS fiberi 1 or 2 cm.

Line 180: Identification took place by using on only one HP-5MS Column capillary column. For proper verification of identification it is necessary to compare retention indices on two columns of different polarity. In most cases substances, which co-elute on one column can be distinguished on the other column with different polarity. Please explain.

Line 189: Is it enough only NSIT library for identification?

Line 190: Authors mentioned that relative contents of all components were calculated by the normalization method. Please explain the normalization method. The concentration of volatile compounds shown in Tables are by reference to single IS including 1,2-o-dichloro- benzene. Is it enough single IS for all compounds? What is the recovery yield of the internal standards? Please explain.

Reviewer 2 Report

Comments and Suggestions for Authors

The manuscript explores the mechanism by which characteristic aromas are produced in Camellia oleifera oil. This topic is relevant to the food industry, but it contains significant methodological inconsistencies and data presentation errors.

  • The most critical issue is the inconsistency in processing temperatures. Camellia seeds were roasted at 170°C (Section 2.2), yet the Maillard reaction and lipid oxidation model systems were heated at 150°C (Section 2.6). Reaction kinetics and the profiles of volatile compounds are highly temperature dependent. Comparing volatiles formed at 150°C in a model system to those formed at 170°C in seeds is methodologically unsound. The authors must explain this choice or explicitly discuss it as a major limitation.
  • The model reactions were carried out in a phosphate buffer (liquid phase), whereas seed roasting is a low-moisture process (solid phase). Water activity fundamentally alters the pathway of the Maillard reaction. This limitation is not adequately discussed; the authors should add a section comparing liquid-state models with solid-state food matrices.
  • In Line 283 and Table 2, there is a significant calculation or unit error. The text states that lysine content decreased by "0.79%." However, based on the data (0.47 to 0.10 mg/g), the actual decrease is approximately 79%.
  • In the Introduction (Lines 58-60), the phrase "...small amount of polar functional groups with good non-polar properties..." is chemically contradictory and should be revised.
  • Table 3 is redundant as it merely presents ratios calculated from the data already shown in Table 2. It should be removed.
  • The authors state that aldehydes (Line 229) decreased due to "excessive heating." Please clarify if this is due to chemical degradation or physical volatilization, given the high temperature (170°C).
  • The manuscript claims lipid oxidation produces heterocycles. While oxygen-containing heterocycles (furans) are expected, nitrogen-containing heterocycles (pyrazines/pyrroles) require a nitrogen source. If System 8 contained only oil and buffer, please explain the origin of nitrogen-containing compounds, if any were detected.
  • Tables 1 and 5 are extremely long. Consider moving the full lists to Supplementary Material and retaining only the key aroma-active compounds in the main text.

Comments on the Quality of English Language

Although the manuscript is generally understandable, the English requires significant polishing to meet publication standards. Awkward phrasing, non-idiomatic sentence structures, and grammatical errors frequently obscure the scientific meaning.

Reviewer 3 Report

Comments and Suggestions for Authors

The article "Formation of characteristic aroma in roasted Camellia oleifera seeds" addresses an important topic: the formation of volatile compounds responsible for the characteristic aroma of raw materials, ingredients, or foods subjected to thermal processes to improve their sensory properties. The results presented on the changes generated during the heat treatment of Camellia oleifera seeds and on the reaction model systems used to determine their formation pathways are very important. However, the authors failed to fully exploit the potential of these results to fully present all the information and provide additional data to the scientific community. Therefore, the following suggestions are provided to improve the manuscript's quality.

The lines relating to comments are indicated with "L" and are also highlighted in the PDF file of the article for reference.

Suggestions and Comments

L155: Standardize the use of periods or commas in concentration figures.

In section 2.3, include a brief description of the quantification of free amino acids.

L162: ¿How was the camellia seed powder defatted? Briefly describe this process or provide a bibliographic reference that describes how the camellia seed powder was defatted.

L168: State the column dimensions. In this section (2.4), indicate how the soluble sugars were quantified.

L180: State column dimensions.

L188: Verify the unit of the mass range. What does "u" mean?

L190-L191: In the Compound Tables, the authors report concentrations in mg/kg; indicate how the quantification was performed, preferably in this section (2.5). What was the concentration of the internal standard solution, and how many µL were added to the headspace vial containing the sample?

If the quantification procedure is described here, the highlighted sentence in the PDF (lines 267-268) can be removed.

L199 and L201: Was the oil used in these systems extracted from unroasted camellia seeds? Please indicate.

L204-205: What weight or volume of the solution from each system was added to the headspace vial for the HS-SPME-GC/MS analysis? This paragraph can be used to describe how the volatile flavor compounds were quantified in the reaction model systems, and the highlighted section in the PDF (lines 457-558) can be removed.

L208: What about PLS-DA? Why is its use not mentioned? L209: Indicate which method of mean comparison was used.

L213: Reformulate the title of subsection 3.1.

Section 3.3: Which individual volatile compounds were strongly correlated with free amino acids according to the PLSR analysis?

L386: Indicate which compounds specifically, according to the PLSR analysis.

Section 3.7: It would be interesting for the authors to include a discussion on which individual volatile compounds formed in the simulated reaction systems coincided with those detected in the oils of roasted camellia seeds at different times. The above would allow us to determine which specific compounds in the complex seed matrix are predominantly generated via which interaction mechanisms.

L418: Write the number 47 at the beginning of the sentence in words to avoid confusion. Or change the position of "Table 5" in the previous sentence so that the numbers 5 and 47 are not together.

L432: In Materials and Methods, it is unclear whether the Maillard-Lipid Oxidation reaction system was carried out with the addition of oxidized camellia oil to the lysine-glucose solution. If the experiment was performed in this way, the authors should describe it clearly in Lines 201-203. As this is not indicated, it is understood that the interaction was with oil extracted by cold pressing of unroasted seeds, as would have occurred in the camellia seed matrix undergoing a roasting process at 170 °C for a set time.

453: It is suggested that the title of Table 5 be changed to something like "Volatile components generated in the different simulated reaction systems."

Only eight esters are reported in Table 5, not 9, as indicated at the beginning of the list for this group of compounds. Please verify.

Also, update the compound numbering: there should be 80 instead of 81. Note that there is a jump in the numbering from 4 to 6. Therefore, L466 should be corrected as well.

L457-458: Rephrase; it is unclear.

It would be much better to describe in the Materials and Methods section how the quantification of volatile compounds was performed, indicating the concentration of the internal standard (IS) solution, how many µL of this IS solution were added to the SPME vial, and what its final concentration was in the matrix, so that it is clearer for the reader.

General Recommendation

-The authors should delve deeper into the analysis and discussion of the results. The article's Tables and Figures contain information that is not addressed or is only discussed superficially.

-The correlation results obtained using the PLSR and PLS-DA tools should be discussed more extensively. In this regard, it is recommended to include supplementary material showing the correlation coefficients between the volatile compounds responsible for the aroma of CO and free amino acids and soluble sugars. It would be interesting if the discussion not only addressed the different functional groups in general terms, but also indicated specific compounds with a high correlation (positive or negative) with the precursors.

-Especially in Section 3.7, it is recommended to expand the description and discussion of the results. Generally speaking, this same information can be inferred from Figure 3. The authors provide valuable information in Table 5 that can be discussed in several ways, focusing on the individual volatile components that were predominantly generated (considering their concentration) in the different reaction systems evaluated.

-The authors mention the characteristic flavor compounds of camellia oil on several occasions. Which compounds are considered key in contributing to the distinctive aroma of camellia oil according to their threshold? (If the authors did not determine this in their process, this information is available in the literature).

Were these key compounds generated during the heat treatment in the reaction model systems and in the complete camellia seed matrix? How many and which ones were matched?

-The authors were able to evaluate the individual contribution of the compounds generated during roasting in both the simulated reaction systems and the seed, and determine which were key compounds under the process conditions using Odor Activity Value Analysis (OAV).

-Question: Why, if sucrose is the main sugar in the control camellia seed matrix, was its use omitted from the model reaction system? While this system is obviously more complex, it would better reflect the nature of the heat-treated seed and the aroma-related compounds that could be generated.

In summary, the authors are encouraged to carefully review the results obtained and explore them from various perspectives, facilitating a better understanding of their significant findings.

Comments on the Quality of English Language

There are some minor aspects of the document's writing that need review.

Reviewer 4 Report

Comments and Suggestions for Authors

Comment 1: Please define a specified membrane filter

supernatant was filtered through a 0.22 µm membrane filter. Using ninhydrin as the deri-          149

Comment 2: Please define a PEEK col žumns

ultraviolet spectrophotometer detector and PEEK col-                                                                         152

umns (4.6 mm × 150 mm). The injection volume was 50 µL, with the column oven main-              153

 Comment 3: Please define a gradient elution

The mobile phase consisted of A (0.04 M trisodium citrate dihydrate and                                          154 0,03 M citric acid, pH 3.5), B (0.07 M trisodium citrate dihydrate and 0.08 M boric acid, pH 155 10) and D (0.2 mg/mL ethylenediaminetetraacetic acid) using a gradient elution. The flow                   156 rate of the elution pump was 0.45 mL/min, and that of the derivatization pump was 0.2                 157 mL/min

Comment 4: Detection limit is necessary to express in mg/g defatted camellia seeds.

The minimum detection limit is 0.002 mg/mL.                                                                             158

Table 2. Changes in free amino acid content in camellia seeds during different roasting time (mg/g  290 defatted camellia seeds).

Comment 5: Detection limit is necessary to express.

2.4. Determination of soluble sugar                                                                 159

The work provides in-depth insight into the complex molecular mechanisms that contribute to the characteristic aroma of roasted Camellia oleifera seeds. Identification of specific volatile compounds and their precursors (such as free amino acids and soluble sugars) is crucial and well described for understanding the process of aroma formation during frying. This knowledge is fundamental to the control and manipulation of taste. By understanding how roasting duration affects the aroma profile and which precursors are key, the authors have provided valuable information so that producers can optimize the roasting process to achieve desired flavor profiles, improve product quality, and develop "fragrant" camellia oil.

In summary, this work is not only important for the specific camellia oil industry, but also for the broader understanding of flavor chemistry in food products, offering scientific guidance for process optimization and product quality improvement.

Round 2

Reviewer 1 Report

Comments and Suggestions for Authors

I reviewed the article titled “Formation of characteristic aroma in roasted Camellia oleifera seeds" again as a referee. The researchers have made necessary corrections, but it needs only minor revisions. Therefore, I can recommend the manuscript for publication in the Foods.

The corrections to be made are detailed below.

Line 252: Add a reference for: (E)-2-heptenal 252 was primarily generated through the oxidative degradation of linoleic acid.

In Table 1: Please delete: (ten kinds), (four kinds) and the others in aroma groups. The number is clear.

In Table 3: Please give two decimal for 86.2±1.51a like the other compounds.

In Table 4: Please delete: (four kinds), (nineteen kinds) and the others in aroma groups. The number is clear.

Reviewer 2 Report

Comments and Suggestions for Authors

The revisions made have significantly improved the quality of the manuscript. I have no further comments on this version and recommend accepting the article.

Comments on the Quality of English Language

Although the manuscript is generally understandable, the English requires significant polishing to meet publication standards. Awkward phrasing, non-idiomatic sentence structures, and grammatical errors frequently obscure the scientific meaning.

Author Response

Dear reviewer,

We sincerely thank you for your further assistance and comments on our manuscript. The valuable suggestions have significantly enhanced the quality of our manuscripts. We apologize for the issues with the English expression in our current manuscript. Therefore, we have subjected the entire document to professional review and meticulous revision. We sought assistance of several researchers with extensive experience in English writing to refine the language sentence by sentence, focusing on correcting grammatical errors, optimizing phrasing, and adjusting non-idiomatic sentence structures. Based on this, we conducted multiple rounds of proofreading to ensure that the language expression is accurate, concise, and consistent throughout. Please refer to the revised manuscript for the specific modifications. All the modified parts are marked in blue. We have tried our best to make all the revisions and hope that the revised manuscript can meet the requirements for publication.

Thanks again!

Reviewer 3 Report

Comments and Suggestions for Authors

The authors have satisfactorily addressed the suggestions and comments on the original version of their manuscript through a cover letter that includes comprehensive, clear, and kind responses. Therefore, the revised version (foods-4004824-peer-review-v2) contains substantial improvements.

However, some aspects still require review. It is requested that the authors again address the suggestions described below and, additionally, consider the comment attached to the suggestion in L219 if it is possible to clarify this aspect of its description in the current manuscript or future studies or publications.

 Suggestions and comments (See PDF file)

-L167: Indicate the reference for the Standard or method mentioned in this section. Add it to the References section. The authors must ensure that the bibliography numbering is correctly reordered throughout the entire document.

-L195: Review the unit of column film thickness, which should be µm (micrometers), not millimeters.

-L202: The authors are correct. In Western countries, it is more common to see these mass ranges indicated as m/z or amu (atomic mass unit). Thank you for the detailed clarification.

-L219: Was the cold-pressed camellia oil used as a blank control the same one used in the lipid oxidation and Maillard-lipid oxidation interaction model system? If so, although it may seem obvious and repetitive, it is also important to indicate "unroasted cold-pressed camellia oil." This way, readers will understand that the oil was provided by Qiyunshan Camellia Science and Technology Co., Ltd., as referenced in L121.

In fact, it is somewhat strange to read that the authors used unroasted, cold-pressed camellia oil obtained from a supplier in the activities described in section 2.6. To accurately evaluate and compare the changes in oil composition during both the roasting of camellia seeds and in the model reaction systems, an alternative strategy would have been to also obtain oil from the unroasted seeds used as a control group (described in L134), i.e., oil obtained from the same raw material without roasting. If crude oil was obtained from roasted seeds by cold pressing with a small press machine, I believe it would also be possible to obtain crude oil from unroasted seeds in the laboratory. The last is important because, without evidence, it is uncertain to know whether the seed batch collected from Deyi Yuan Ecological Agriculture Development Co., Ltd. and the seeds used to produce the "unroasted cold-pressed camellia oil" provided by Qiyunshan Camellia Science and Technology Co., Ltd. had the same initial composition.

For example, in Table 1, was the volatile composition of camellia oil at time = 0 min either obtained from unroasted cold-pressed camellia oil provided by the company (L121) or from unroasted seeds at the laboratory or pilot level by the authors? When the authors describe unroasted camellia seeds used as the control group (L134), it is unclear whether they are referring to this control group solely for obtaining the fine powder (for the determination of free amino acids and soluble sugars) or also for obtaining crude oil from unroasted camellia seeds. The above, considering that everything could be evaluated using the same batch of camellia seeds.

-L425: Please complete the name of the compound.

-L545: If the authors consider that referring to “concentration of precursor compounds” or “initial reactant concentration” is equivalent to “local reactant concentration,” it is suggested that this be changed.
